# RJA-Star Algorithm for UAV Path Planning Based on Improved R5DOS Model

Jian Li [1,2], Weijian Zhang [1], Yating Hu [1,2,*], Shengliang Fu [1], Changyi Liao [1] and Weilin Yu [1]

1   College of Information Technology, Jilin Agricultural University, Changchun 130118, China
2   Bioinformatics Research Center of Jilin Province, Changchun 130118, China
*   Correspondence: huyating@jlau.edu.cn; Tel.: +86-178-3312-5736

**Abstract:** To improve the obstacle avoidance ability of agricultural unmanned aerial vehicles (UAV) in farmland settings, a three-dimensional space path planning model based on the R5DOS model is proposed in this paper. The direction layer of the R5DOS intersection model is improved, and the RJA-star algorithm is constructed with the improved jump point search A-star algorithm in our paper. The R5DOS model is simulated in MATLAB. The simulation results show that this model can reduce the computational complexity, computation time, the number of corners and the maximum angles of the A-star algorithm. Compared with the traditional algorithm, the model can avoid obstacles effectively and reduce the reaction times of the UAV. The final fitting results show that compared with A-star algorithm, the RJA-star algorithm reduced the total distance by 2.53%, the computation time by 97.65%, the number of nodes by 99.96% and the number of corners by 96.08% with the maximum corners reduced by approximately 63.30%. Compared with the geometric A-star algorithm, the running time of the RJA-star algorithm is reduced by 95.84%, the number of nodes is reduced by 99.95%, and the number of turns is reduced by 67.28%. In general, the experimental results confirm the effectiveness and feasibility of RJA star algorithm in three-dimensional space obstacle avoidance.

**Keywords:** R5DOS intersection matrix; RJA-star algorithm; jump point search algorithm; path planning





## 1. Introduction

Smart agriculture means that rural workers use advanced technology and experience to carry out agricultural production. The development of social, economic and technological smart agriculture in rural areas is closely related [1]. which is essential in eliminating poverty, helping developed economies "catch up" and forming strategies that promote development in China. Intelligent agriculture includes areas such as: internet technologies, wireless sensor technology [2,3] and remote control technology. As a very important part of intelligent equipment, UAVs are the focus of more attention because of their practical usefulness. UAVs are widely used in agriculture [4,5], forestry [6], disaster relief [7], and geological exploration [8]. The Agricultural UAV [9,10] plays an important role in the fields of crop monitoring, crop yield assessment and plant protection. However, when the UAV sprays precisely at low altitude, farmland obstacles such as plant protection network, residence, electric pole, communication tower, lighting objects and various organisms will pose a serious threat to the UAVs [11]. The main task in agricultural work is to effectively avoid obstacles and achieve the set goals. Significant research has been carried out to solve this problem. Autonomous flight can be achieved to a certain extent by means of sensors, reliable control algorithms and pre-measured obstacle position information [12,13].

A large amount of path planning and obstacle avoidance algorithms have been developed for the path planning of UAVs. For example, A-star algorithm [14], ant colony algorithm [15], artificial potential field [16], DIJKSTRA [17] and so on. Deep learning technology can also be used as an effective method for UAV path planning. A-star algorithm is an effective tool, but the computational complexity of the traditional A-star algorithm

space will increase exponentially with the size of the map, resulting in a significant increase in computational complexity and computational time. In order to solve these problems, domestic and foreign scholars have done significant research. Anh et al. [18]., proposed a zigzag global planner based on a-star, which improves the search efficiency of a-star algorithm in narrow space. Hong et al. [19]., proposed an improved A-star algorithm based on terrain data, reducing the calculation time of the algorithm. Zhang et al. [20]., added A-star algorithm to the artificial potential field algorithm, which optimizes the path xu of the UAV and better reflects the actual environment of the UAV. Li et al. [21]., improved the genetic algorithm with A-star algorithm. Ma et al. [22]., adopted the idea of collision ji to improve A-star algorithm. Li et al. [23]., proposed an improved A-star algorithm combined with the jump point search algorithm to reduce the computational cost of A-star algorithm. However, there are still problems in using the nearest neighbor interpolation method to process the path.

To sum up, the a-star algorithm in two-dimensional space has been greatly improved. But there is little improvement regarding the three-dimensional space A-star algorithm [24]. Dilip Mandloi et al. [25], made a detailed comparison on the path planning strategies of star a, Lazy Theta and Theta in a three-dimensional environment. Zhang et al. [26], constructed an improved path planning model using three-dimensional two-way sector expansion method and variable step search strategy. Although the moving distance is reduced by 7.53%, the time cost is increased by 2.66 times. These researches still have the problems of complex computation and large time cost, to solve this problem, this paper presents an algorithm of RJA-star (R5DOS Jump A-star) based on the R5DOS(RCC5-Direction-Octant-Strongly-exists model) model in three-dimensional space. Firstly, the topology of UAV and UAV detection area is expressed, and the A-star algorithm is improved by the JPS(jump point search) algorithm. The model combines the JPS algorithm with the three-dimensional space algorithm to improve A-star algorithm.

The purpose of this study is as follows:

(1) Reduce the complexity of A-star algorithm in three-dimensional space.
(2) Optimize the path selection of A-star algorithm, reduce the corners of the path and make the path more smooth.
(3) Reduce the mobile cost of UAVs, including computing time, path length and access nodes.

## 2. Materials and Methods

### 2.1. Abstract Topological Representation of UAV

Li et al., improved the R5DOS model and proposed a multi UAV formation model, dividing the space into 16 regions, combined with Topology [27]. We modified the R5DOS model, cancelled its formation, and placed the UAV at the center of the model.

According to the R5DOS model, the UAV can be divided into two parts: the body area and the detection area. The detection area is used as the area for UAV to obtain information and perceive the surrounding environment, and is mainly responsible for detecting obstacles and target points, this can be considered a safe area. UAV fuselage area is the area where obstacles need to be avoided during flight.

According to reference [28], we can get the definition of five topological relationships, which are: Discrete (DR), Partial Overlap (PO), Proper Part (PP), Equal (EQ), Proper Part Inverses (PPI), as shown in Figure 1.

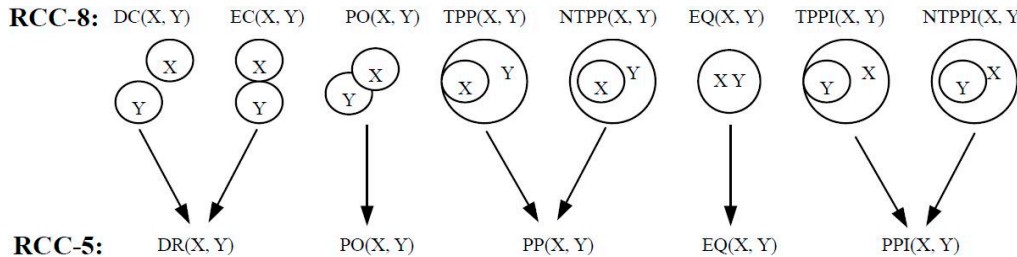

**Figure 1.** Definition and representation of five topological relationships.

The UAV Area C is included in area B, satisfies topological relationship PP (B, C), note that the coordinates of the UAV are $(x_u, y_u, z_u)$, and the coordinates of the target point are $(x_t, y_t, z_t)$, as shown in Figure 2:

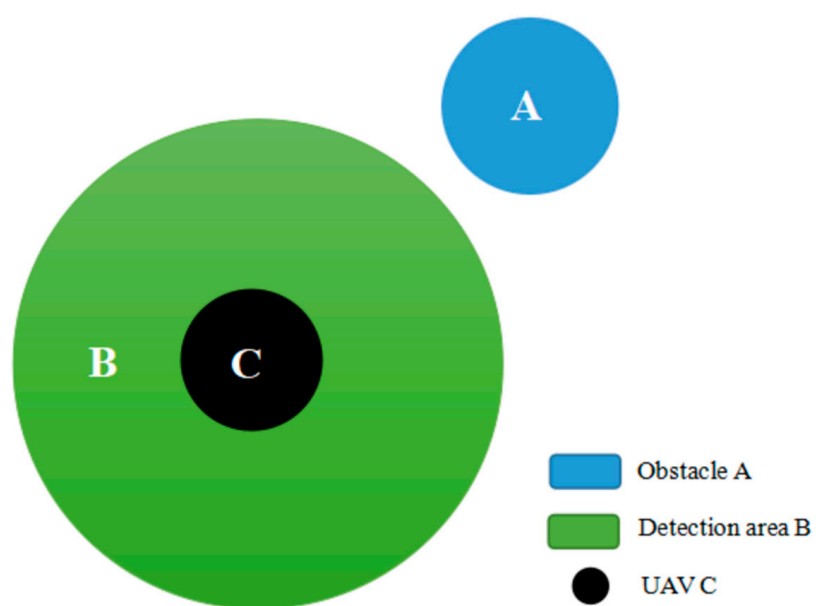

**Figure 2.** Abstract topology representation of UAV area.

### 2.2. Brief Introduction to A-Star Algorithm

The A-star algorithm is a method of finding the shortest path in static scenes. The calculation principle of the A-star algorithm is both simple and fast, but this will slow down with the increase of map size. The increase in computing space will also be exponential, with the larger the map, the greater the computing time and storage footprint. The A-star algorithm will take significant time and space for path calculation. In addition, the computation time will determine the routing efficiency of A star algorithm.

The A-star algorithm is a fast and efficient way to find the right heuristic. Among them, $H(n)$ is the function expression of the shortest path cost of UAV starting from the nth node $(x_n, y_n, z_n)$ to the target $(x_t, y_t, z_t)$. $G(n)$ represents the shortest path between the starting point $(x_s, y_s, z_s)$ and the nth node. Wherein, $F_{cot}(n)$, $H(n)$, $G(n)$ can be expressed by Equation (1).

$$
\begin{aligned}
H(n) &= \sqrt{(x_t - x_n)^2 + (y_t - y_n)^2 + (z_t - z_n)^2} \\
G(n) &= \sqrt{(x_n - x_s)^2 + (y_n - y_s)^2 + (z_n - z_s)^2} \\
F_{cot}(n) &= G(n) + H(n)
\end{aligned}
\tag{1}
$$

### 2.3. Jump Point Search Algorithms

A-star requires a lot of unnecessary computation and memory space [29], to improve this an adaptive JPS algorithm was introduced. JPS algorithm is a special search strategy, which only accesses special nodes in the calculation process [30]. This can be thought of as a pre-processing method. After pre-processing, the remaining nodes to be searched are called jump points. As shown in Figure 3, the current node is dotted in blue. From the blue node A5 to the orange node G5, we can ignore the grey node B4, C4, B6, C6 and so on, because the cost of going through B5 to G5 is minimal. When extending to node G5, since node G6 is an obstacle, G5 is the forced neighbor of hop G6.

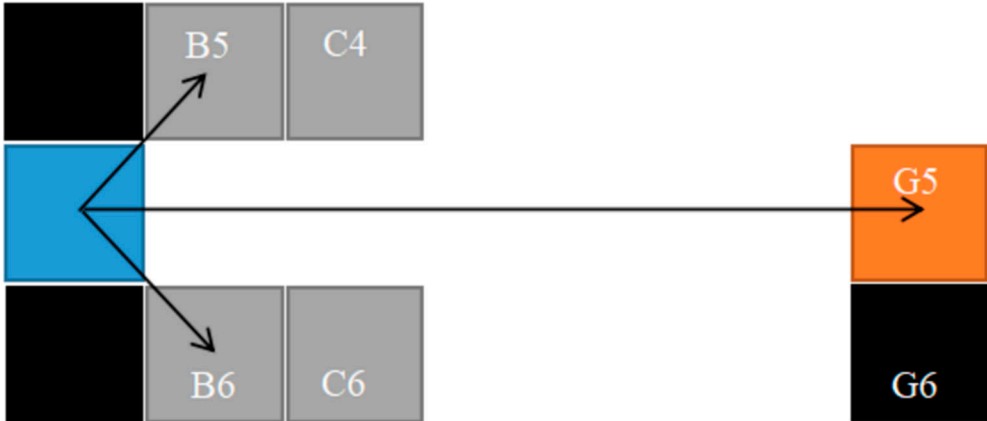

**Figure 3.** The shortest path selection example of JPS algorithm.

This will greatly increase computing time while reducing computing efficiency. In this paper, we will add a preprocessing process to all nodes based on the A-star algorithm to obtain a batch of special hops. In this way, the improved RJA star algorithm can reduce a lot of unnecessary analysis and calculation.

This article defines it according to the actual situation. Take the three-dimensional view of Figure 4a and the cross-section of Figure 4b as an example, assuming that the grey area is an obstacle. The connecting line between UAV and target is defined as $L_{ut}$, the length as $d_{ut} = \sqrt{(x_t - x_u)^2 + (y_t - y_u)^2 + (z_t - z_u)^2}$, and the direction vector as $\vec{L}_{ut} = \left( \frac{(x_t - x_u)}{d_{ut}}, \frac{(y_t - y_u)}{d_{ut}}, \frac{(z_t - z_u)}{d_{ut}} \right)$. Taking the connection line $L_{ut}$ as the central axis of the detection area B, the length of the UAV body is r, and in order to ensure that the UAV can safely avoid obstacles without being affected by other factors, we set the diameter of B as 3r, to determine if there is an obstacle $A_i, (i = 1, 2, \cdots)$ and B intersecting, that is, the topological relationship is $R(A_i, B, C) = \begin{pmatrix} 0 & 1 & 0 & 1 \\ 1 & 1 & 0 & 1 \end{pmatrix}$, where B represents the detection area and C represents the UAV body Area.

If such an obstacle exists, the nearest obstacle to the UAV is considered as the forcing neighbor, the vertices and boundary points with special distance on the obstacle are regarded as the next round of search nodes, namely, that is, the yellow nodes in the Figure 4a,b.

When the algorithm executes the JPS algorithm once, it repeats the above steps until no obstacle intersects with $L_{ut}$ in Figure 5b, and then completes the JPS. Note:

$$\begin{cases} J_0, & \text{the coordinates of the starting point of the UAV} \\ J_i (i = 1, 2, \cdots, n-1), & \text{the coordinates of the jump point} \\ J_n, & \text{the coordinates of the target point} \end{cases}$$

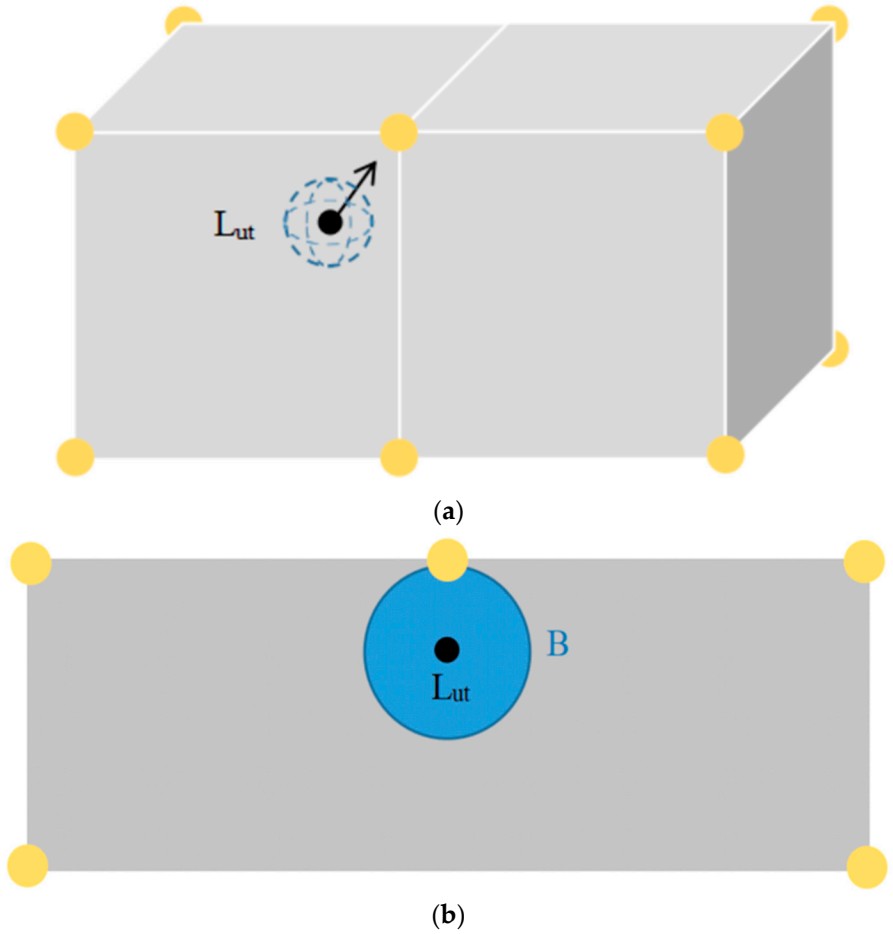

**(a)**

**(b)**

**Figure 4.** (**a**) Schematic diagram of obstacles and UAV. (**b**) Cross sectional sketch of obstacles and UAV.

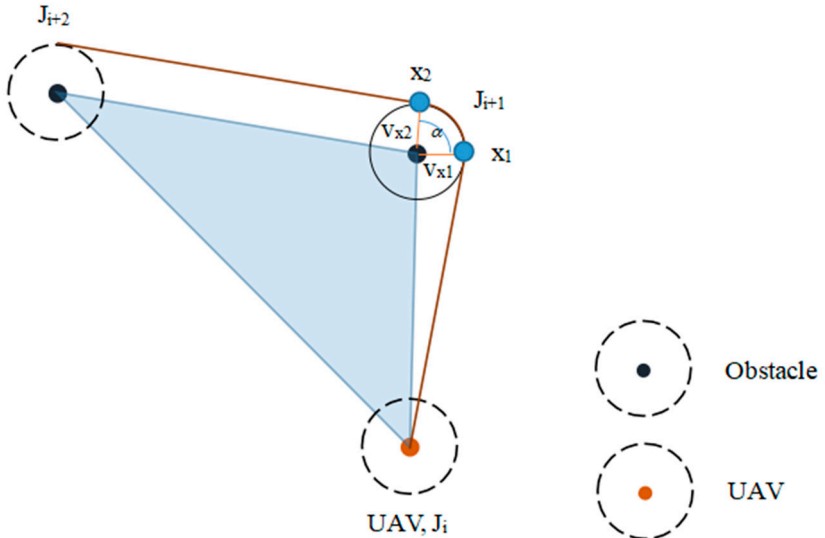

**Figure 5.** Path Selection for UAV Obstacle Avoidance.

During the implementation of the A-star algorithm, the path is not smooth enough because there are many corners, which does not conform to the movement mode of the UAV. To ensure that the UAV can move smoothly and be safe enough, first, we filter out all the jumping points. Then consider expanding a sphere with a diameter of 3r at each node and call it a "jumping body". At the same time, the moving target of UAV is not the selected

jumping point, but the jumping body corresponding to each jumping point. Starting from the origin, every time the UAV moves to the next jump body, we will obtain the coordinates of the current node and the next two jump points of the UAV. In the three-dimensional plane the two jumping bodies will intersect on the corresponding spherical tangent plane.

There are two types of UAV paths:

1. When flying from the starting point, move the tangent line from the current position to the circle.
2. Obtain two tangent points $x_1$ and $x_2$ of the same circle on the path, connect them with the center of the circle to obtain the vectors $V_{x1}$ and $V_{x2}$, then the rotation angle is: $\cos(\alpha) = \frac{V_{x1} \bullet V_{x2}}{|V_{x1}| \bullet |V_{x2}|}$.
3. When moving from one body to another, first move along the circle until you leave the body, then move toward the tangent of the next body, as shown in Figure 5.

The current position of the UAV is updated after each movement, and the process is repeated until the movement reaches the target point.

### 2.4. Improved Path Planning Algorithm

According to Sections 2.1–2.4, RJA star path planning Algorithm 1 is as follows:

1. Detects obstacles in the path between the UAV current position and the target point.
2. If there is an obstacle, consider the one closest to the UAV as a coercive neighbor. If there is no obstacle, end the search and fly directly to the target point.
3. Consider the Vertex of the forcing neighbor and its closest boundary point as the jump point.
4. Repeat steps 2–3 until all hops are searched.
5. Compute the cost functions for all hops, compare them, and choose the best path.
6. The UAV chooses the jumping body to move in turn, and the moving path is on the plane formed by the adjacent jumping points.
7. Move in sequence until you reach the target point.

The pseudo-code of RJA-star is:

---

**Algorithm 1: RJA-star**

---

1　algorithm RJA-star (start point, current_node, child node, goal point)
2　　$F_{cot}$(current_node) = G(current_node) + H(current_node)
3　if current_node = goal point found the feasible path; break
4　Generate child node that come after current_node
5　for child node of current_node
6　　　if $R(A_i, B, C) = \begin{pmatrix} 0 & 0 & 0 & 1 \\ 1 & 1 & 0 & 1 \end{pmatrix}$ then Jumpbody.f ← Ø
7　　　　if $R(A_i, B, C) = \begin{pmatrix} 0 & 1 & 0 & 1 \\ 1 & 1 & 0 & 1 \end{pmatrix}$ then open ← Jump bodys
8　　　　　else closed ← Jump bodys
9　　　end if
10　　end if
11　　　closed ← current_node
12　　　　if Path(current_node) and RJAPath(current_node) then
13　　　　　final ← closed
14　　　　end if
15 end for
16 end
17 return
18 if (current_node ! = goal point) exit with error (the open list is empty)

---

The algorithm flow is shown in Figure 6.

**Figure 6.** Work flow diagram of RJA star algorithm.

## 3. Results

We simulated maps of different map sizes in Matlab. The starting point is (0,0,0), aand the target point is the point farthest from the origin under the current dimension, and generate disjoint obstacles based on the xoy plane. The number of obstacles is 8/10 of the projected area of the current map on the xoy plane, and then a random height is given.

In order to discuss the advantages of RJA star algorithm more intuitively, we simulate A-star algorithm and RJA star algorithm on maps of different sizes. Calculate the running time of the algorithm to evaluate the effectiveness of the RJA star algorithm.The running time starts from the map generation and ends when the UAV reaches the target point

Our simulation environment is as follows:

CPU:Intel® Core™ i7-8750H;

GPU:NVIDIA GTX 1060 Max-Q 6 GB.

We repeated the average value of 100 simulation experiments as the result. We fixed the height of the experimental map at 15 m and set several scenarios, as shown in Table 1.

**Table 1.** Scenario Names of Different Map Sizes.

| Scenario Name | Map Size | Scenario Name | Map Size |
|---|---|---|---|
| scenario 1 | $20 \times 20 \times 15$ | scenario 7 | $80 \times 80 \times 15$ |
| scenario 2 | $30 \times 30 \times 15$ | scenario 8 | $90 \times 90 \times 15$ |
| scenario 3 | $40 \times 40 \times 15$ | scenario 9 | $100 \times 100 \times 15$ |
| scenario 4 | $50 \times 50 \times 15$ | scenario 10 | $110 \times 110 \times 15$ |
| scenario 5 | $60 \times 60 \times 15$ | scenario 11 | $30 \times 90 \times 15$ |
| scenario 6 | $70 \times 70 \times 15$ | | |

We simulate the RJA star algorithm and A-star algorithm in scenario 11, as shown in Figure 7a–d.

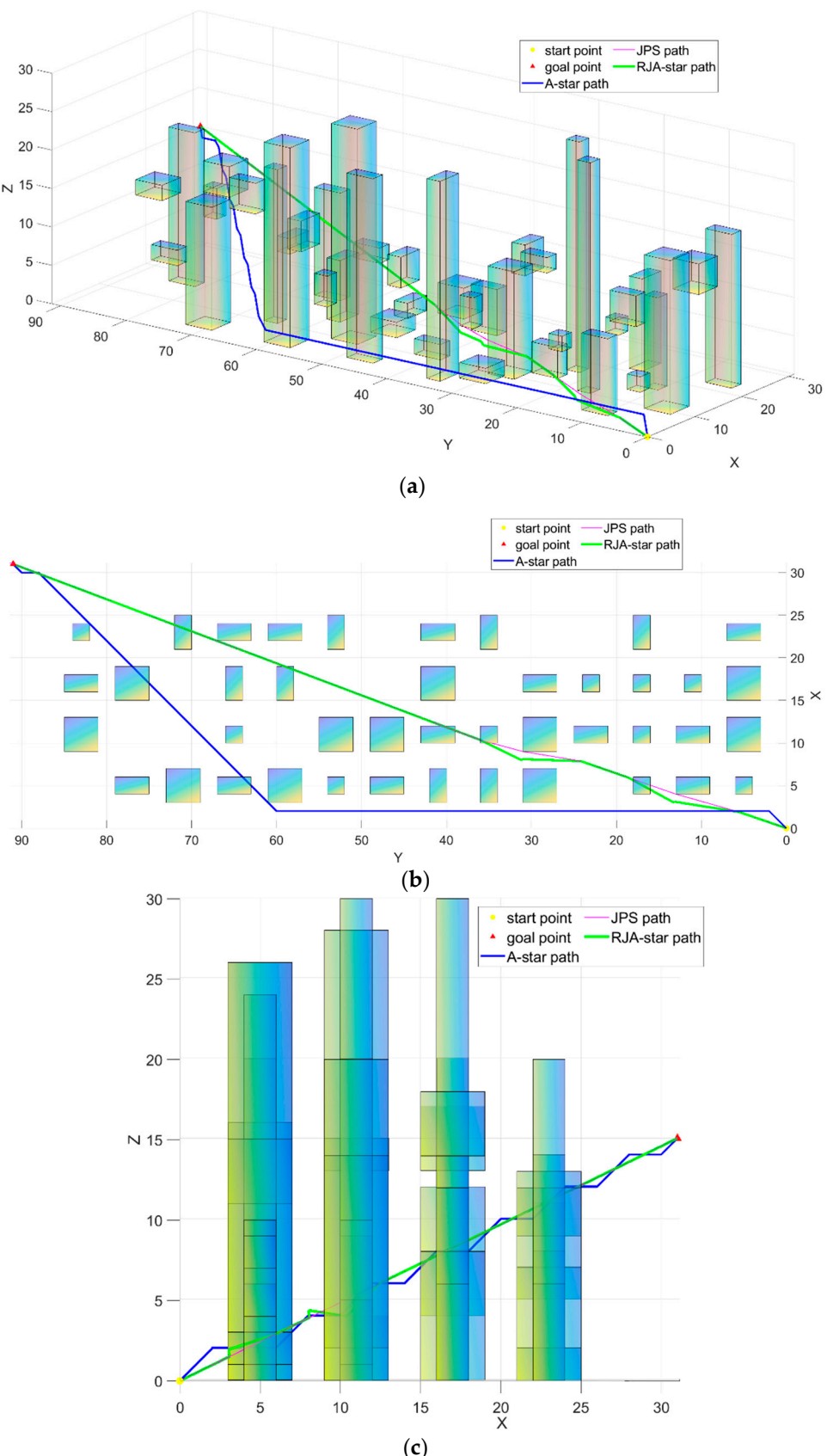

**Figure 7.** *Cont.*

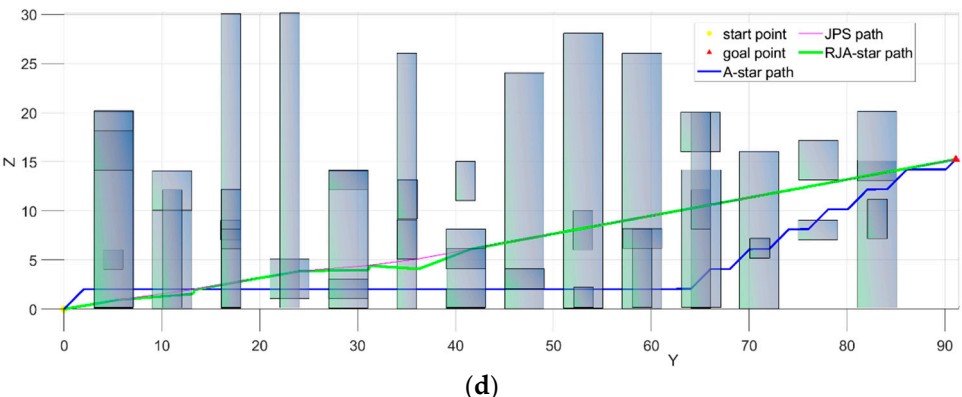

(**d**)

**Figure 7.** (**a**) The fitting results and details of RJA star algorithm on scenario 11. (**b**) Fitting result of XY plane of RJA star algorithm in scenario 11. (**c**) Fitting result of XZ plane of RJA star algorithm in scenario 11. (**d**) Fitting result of YZ plane of RJA star algorithm in scenario 11.

To compare the experimental results, we use Equation (2) to calculate the results of RJA star algorithm and A-star algorithm in computing time, exploration nodes, and path length.

$$\frac{(I_A - I_{RJA})}{I_A} \times 100\%, I_{A/RJA} = \{computing\ time,\ exploration\ nodes, path\ length\} \quad (2)$$

Through Equation (2), we can calculate the reduced computation time, probe node and path length of the RJA star algorithm. Figure 8 is the results of a simulation run under scenario 11. Compared with A-star algorithm, RJA star algorithm reduces 94.4% rotation angle, 99.87% calculation time and 10.05% path length. It can be seen from the results in Figure 9 that the path of the JPS algorithm will stick to the obstacle, which will threaten the safety of the UAV. The RJA star path is smoother and safer than the JPS algorithm, and it can avoid obstacles to reach the destination in many cases. We can see that the RJA-star algorithm avoided obstacles well.

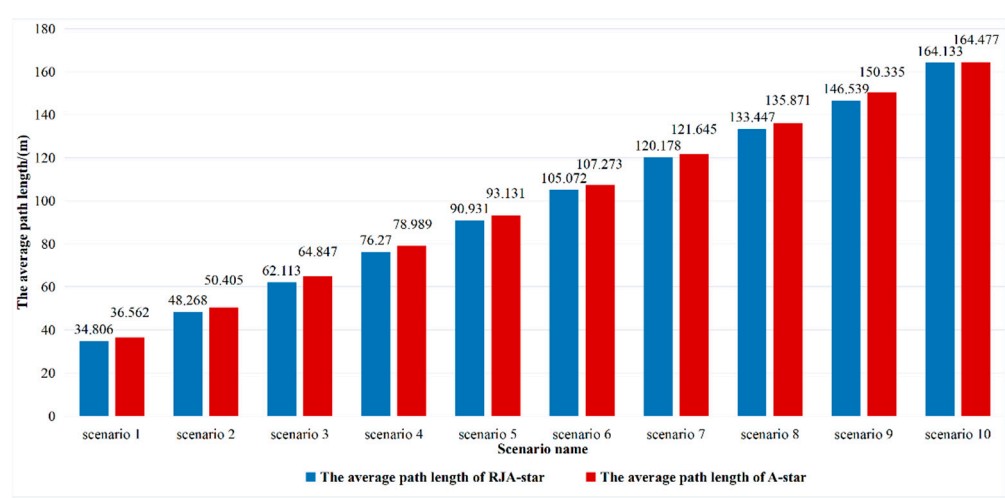

**Figure 8.** Comparison of path length between the two algorithms.

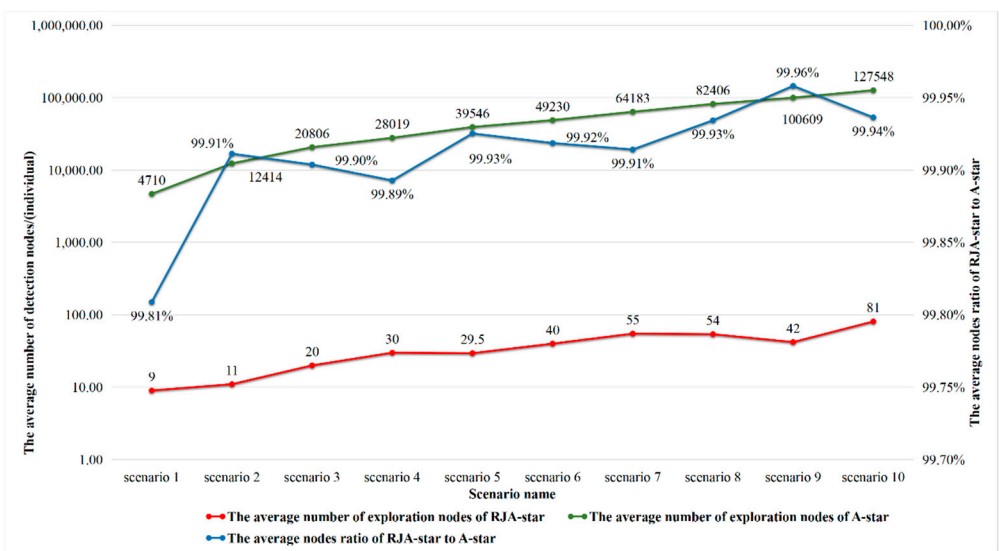

**Figure 9.** Comparison of two algorithms for exploring nodes.

After 100 simulation experiments, we fit the two algorithms into the scenario 1—scenario 10. We limit the random range within the current map, and generate disjoint obstacles based on the xoy plane. The number of obstacles is 8/10 of the projected area of the current map on the xoy plane, and then a random height is given. We compare the two algorithms on a 20 m long map and a 10 m long map. We can obtain the path lengths (as shown in Figure 8), exploration nodes (as shown in Figure 9), computation time (as shown in Figure 10) and the number of corners (as shown in Figure 11) for the A-star algorithm and the RJA-star algorithm. The maximum number of corners are shown in Figure 11.

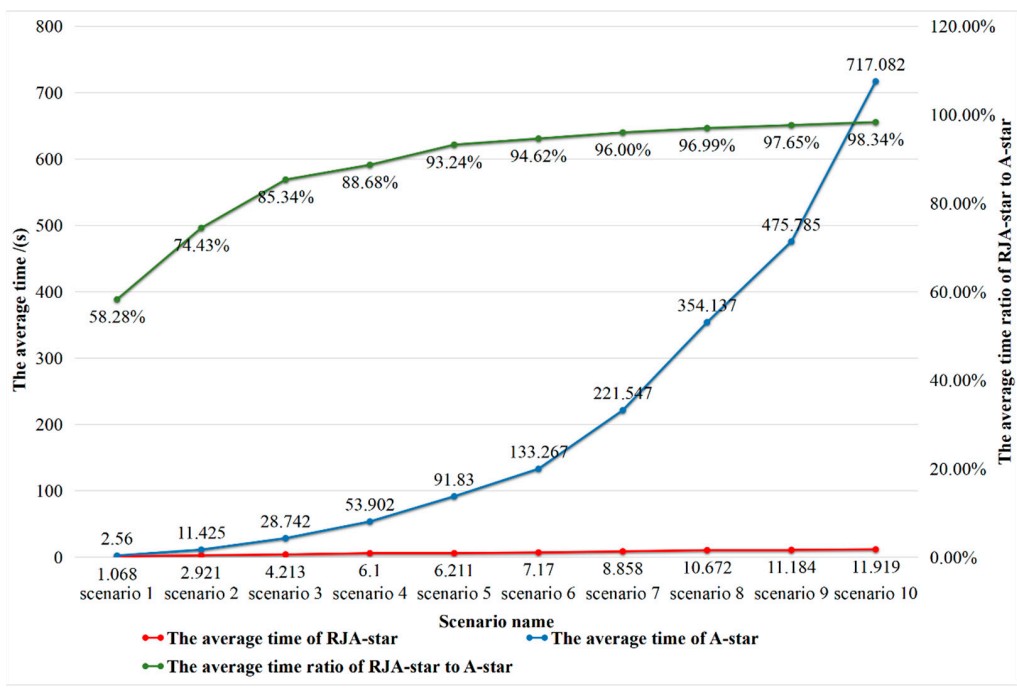

**Figure 10.** Comparison of the average time of the two algorithms.

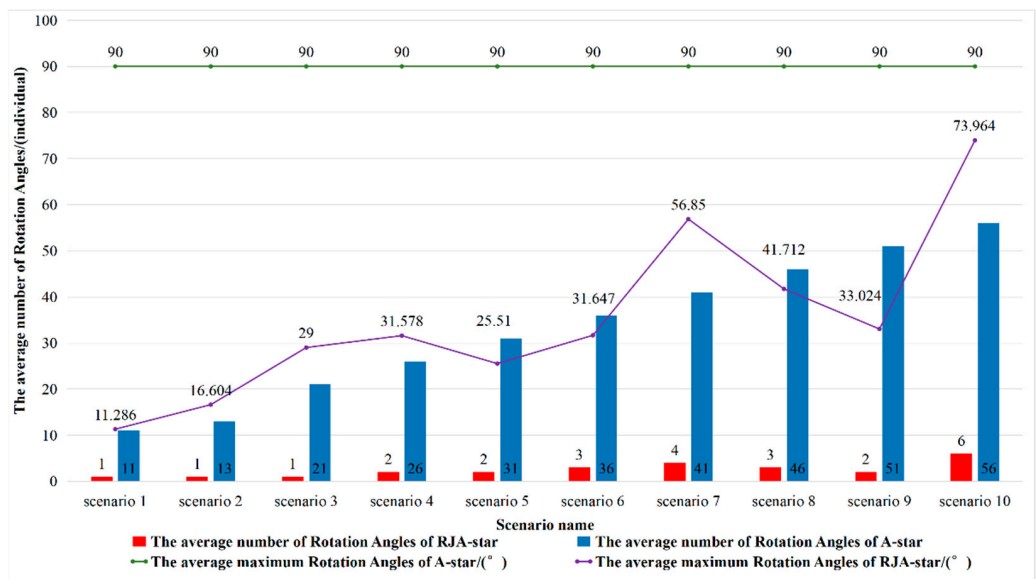

**Figure 11.** Comparison of the number of corners and the maximum angle between the two algorithms.

The result shows that the path length of the RJA star algorithm is not much different from that of the A-star algorithm.

It can be seen from the results in the Figure that the number of nodes of RJA star algorithm is far less than that of A star algorithm.

The y-axis on the left of Figures 9 and 10 respectively represents the number of exploration nodes and average time, and the y-axis on the right represents the percentage reduction of RJA star algorithm compared with A-star algorithm,, which can be calculated by Equation (2). We may know from the Figure that RJA star algorithm has the advantage of short computing time. The five sizes of the map are compared in Table 2.

**Table 2.** Comparison of experimental data of three different specifications of network diagrams.

| Map Size | Path Parameters | Traditional A-Star | RJA-Star | RJA-Star Reduced Proportion |
|---|---|---|---|---|
| | Average running time/s | 2.56 | 1.068 | 58.28% |
| | Average number of nodes | 4710 | 9 | 99.81% |
| 20 × 20 | Total distance/s | 36.806 | 34.806 | 4.80% |
| | Max turning | 90° | 11.286° | 96.59% |
| | Number of turns | 11 | 1 | 87.46% |
| | Average running time/s | 28.742 | 4.213 | 85.34% |
| | Average number of nodes | 20,806 | 20 | 99.90% |
| 40 × 40 | Total distance/s | 64.847 | 62.113 | 4.22% |
| | Max turning | 90° | 29° | 67.78% |
| | Number of turns | 21 | 1 | 95.24% |
| | Average running time/s | 91.83 | 6.211 | 93.24% |
| | Average number of nodes | 39,546 | 29.5 | 99.93% |
| 60 × 60 | Total distance/s | 93.131 | 90.931 | 2.36% |
| | Max turning | 90° | 25.51° | 71.66% |
| | Number of turns | 31 | 2 | 93.55% |

**Table 2.** *Cont.*

| Map Size | Path Parameters | Traditional A-Star | RJA-Star | RJA-Star Reduced Proportion |
|---|---|---|---|---|
| | Average running time/s | 221.547 | 8.858 | 96.00% |
| | Average number of nodes | 64,183 | 55 | 99.91% |
| 80 × 80 | Total distance/s | 121.645 | 120.178 | 1.21% |
| | Max turning | 90° | 56.85° | 36.83% |
| | Number of turns | 41 | 4 | 90.24% |
| | Average running time/s | 475.785 | 11.184 | 97.65% |
| | Average number of nodes | 100,609 | 81 | 99.96% |
| 100 × 100 | Total distance/s | 150.335 | 146.839 | 2.53% |
| | Max turning | 90° | 33.024° | 63.31% |
| | Number of turns | 56 | 2 | 96.08% |

*Discussion*

We select scenario 1, 4, 7, 9. Make the number of obstacles equal to 5/10 of the area of the xoy plane, run the algorithm 20 times on each map, and conduct statistical analysis on the algorithm. From the results, we can get the mean and variance of RJA star algorithm and A-star algorithm under each size map, as shown in Table 3.

**Table 3.** Mean and variance of different indexes of two algorithms.

| Map Size | Map ID | Evaluating Indicator | Estimated Parameters | A-Star | RJA-Star |
|---|---|---|---|---|---|
| | | Average running time/s | mean | 1.194 | 0.501 |
| | | | variance | 0.106 | 0.031 |
| 20 × 20 | 1–20 | Average number of nodes | mean | 1937.950 | 12.300 |
| | | | variance | 3,139,070.892 | 85.484 |
| | | Total distance/s | mean | 34.729 | 34.007 |
| | | | variance | 0.293 | 1.181 |
| | | Average running time/s | mean | 33.634 | 2.900 |
| | | | variance | 619.427 | 0.352 |
| 50 × 50 | 21–40 | Average number of nodes | mean | 23,298.050 | 19.050 |
| | | | variance | 57,624,445.210 | 96.787 |
| | | Total distance/s | mean | 77.010 | 74.192 |
| | | | variance | 0.275 | 0.624 |
| | | Average running time/s | mean | 119.688 | 4.893 |
| | | | variance | 308.581 | 0.844 |
| 50 × 50 | 41–60 | Average number of nodes | mean | 48,848.400 | 37.550 |
| | | | variance | 23,670,949.090 | 350.892 |
| | | Total distance/s | mean | 119.351 | 117.597 |
| | | | variance | 0.020 | 15.461 |
| | | Average running time/s | mean | 323.165 | 2.900 |
| | | | variance | 50,306.815 | 0.352 |
| 100 × 100 | 61–80 | Average number of nodes | mean | 79,975.700 | 40.900 |
| | | | variance | 829,201,538.100 | 526.937 |
| | | Total distance/s | mean | 147.947 | 145.607 |
| | | | variance | 0.454 | 12.160 |

It can be concluded from Table 3 that the RJA star algorithm has smaller variance and mean value in both running time and nodes, which means that the RJA star algorithm is more stable and excellent in running time and nodes performance. The RJA star algorithm considers the safety distance, so the mean value of path is less than A-star algorithm, but the variance is greater than A-star algorithm. Regression analysis was conducted for our data. As shown in Figure 12a–c.

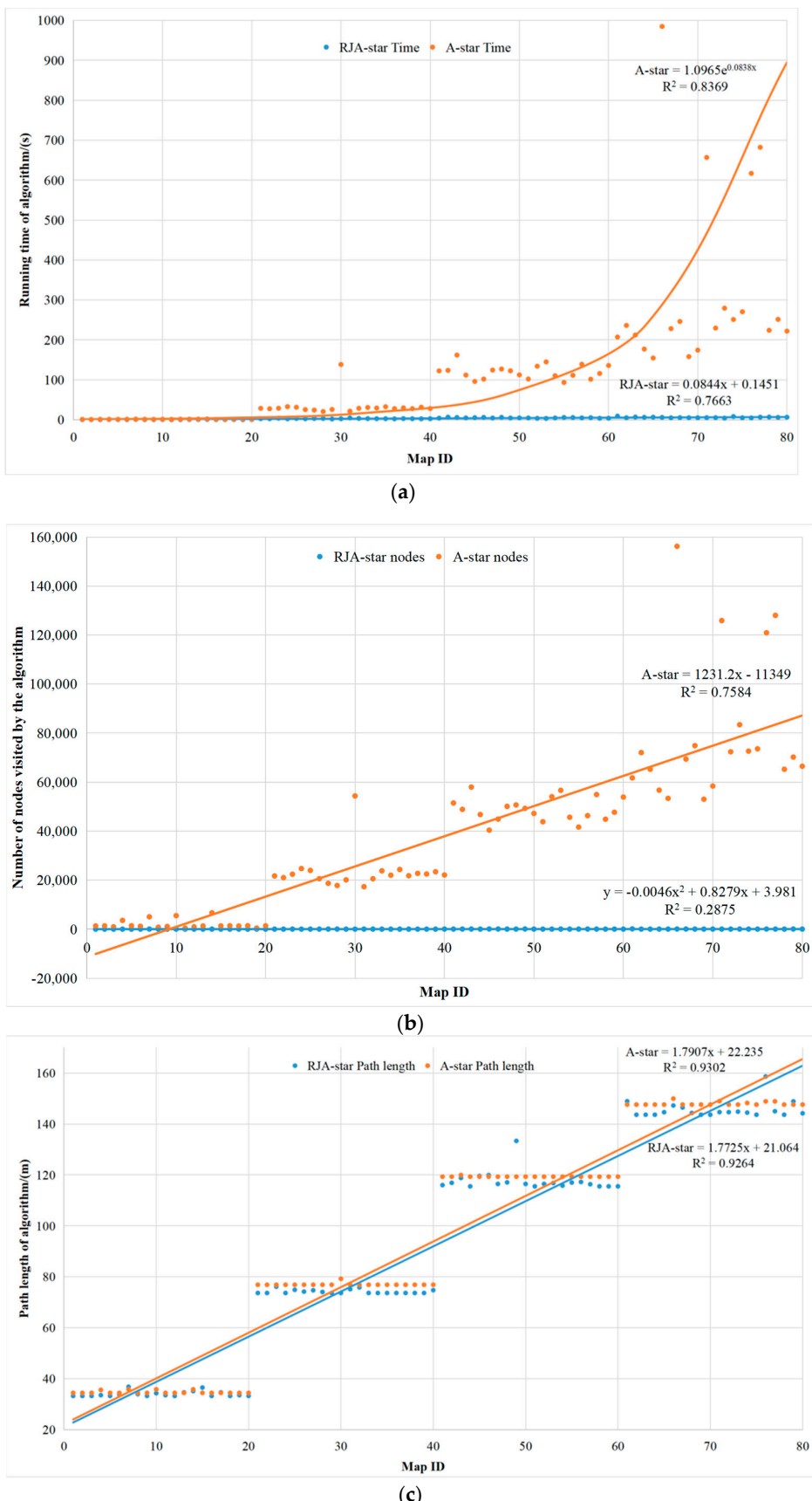

**Figure 12.** (**a**) Regression analysis of computing time of two algorithms. (**b**) Regression analysis of two algorithms' access nodes. (**c**) Regression analysis of path length of two algorithms.

We can see that the number of nodes and path moving length of the two algorithms show a linear regression trend. The regression accuracy of their path lengths can reach 93.02% and 92.64% respectively. The running time of A-star algorithm is exponential regression, with an accuracy of 83.69%. The running time of RJA star algorithm is linear regression, with an accuracy of 76.63%.

Tang et al. proposed the geometric A-star algorithm [31]. To compare the advantages and disadvantages of A-star algorithm, geometric A-star algorithm and improved RJA algorithm, we simulated 100 experiments in scenario 9, and took the average value to Table 4.

**Table 4.** Comparison of indexes of three algorithms.

| Path Parameters | Traditional A-Star | RJA-Star | Geometric A-Star Reduced Proportion | RJA-Star Reduced Proportion |
|---|---|---|---|---|
| Average running time/s | 475.785 | 11.184 | 41.1% | 97.65% |
| Average number of nodes | 100,609 | 81 | 19% | 99.96% |
| Total distance/s | 150.335 | 146.839 | 13% | 2.53% |
| Max turning | 90° | 33.024° | 66.7% | 63.30% |
| Number of turns | 56 | 2 | 84.1% | 96.08% |

Overall, the RJA-star algorithm performs better in terms of computing time and detecting nodes.

## 4. Conclusions

This paper proposes a new path planning algorithm, namely RJA star algorithm. The experimental results show that the RJA star algorithm reduces the mobile path by 2.53%, the computation time by 97.65%, the number of nodes by 99.96% and the number of corners by 96.08% with the maximum corners reduced by approximately 63.30%.smaller angles and smoother paths. In general, our method can effectively reduce the number of turns, calculate the nodes and the angle of turns in the process of UAV motion.

The contributions of this study are as follows:

(1) In order to solve the problem of high computational complexity of three-dimensional A-star algorithm, RJA star path algorithm is proposed.
(2) The pretreatment process is added to screen out a batch of hops and optimize the path between hops. The access and calculation time of nodes are reduced, and the calculation speed is improved.
(3) When obstacles are detected, the algorithm generates hops to optimize the moving path to avoid obstacles.

The organizational structure of this paper is: In the Section 2, we introduce the specific improved form of RJA star algorithm and pseudo code. In Section 3, we simulated in a 3D map to verify the progressiveness of our algorithm. Finally, we draw conclusions in Section 4 and conceive the future work.

The RJA-star algorithm mainly works offline on already known scenarios, without considering the fact that the actual working environment of the UAV should be dynamic and complex. Future work should focus on studying complex dynamic scenarios, including the consideration of flying birds and moving agricultural machinery and other factors.

**Author Contributions:** Conceptualization, J.L. and W.Z.; methodology, J.L. and Y.H.; software, S.F. and C.L.; validation, S.F., C.L. and W.Y.; data curation, C.L.; writing—original draft preparation, W.Z.; writing—review and editing, W.Z.; funding acquisition, J.L. and Y.H. All authors have read and agreed to the published version of the manuscript.

**Funding:** This research was funded by Jilin Province Development and Reform Commission China, grant number 2020C037-7; The Education Department of Jilin Province China, grant number JJKH20220332KJ; The Science and Technology Project of Education Department of Jilin Province grant number JJKH20220330KJ; Changchun Science and Technology Development Plan China, grant number 21ZGN26.

**Data Availability Statement:** Not applicable.

**Conflicts of Interest:** The authors declare no conflict of interest.

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
