# Peer review of "RJA-Star Algorithm for UAV Path Planning Based on Improved R5DOS Model"

_applsci, doi:10.3390/app13021105_

Round 1

Reviewer 1 Report

The authors present an article in which they develop a three-dimensional space path planning algorithm (RJA star) based on the R5DOS model. The authors establish that the proposed algorithm outperforms the A-star algorithm.

The study fits within the scope of the Applied Sciences journal, and it is an interesting work. However, the manuscript could be improved and has some flaws that prevent its publication in the journal.

The objectives of the paper are not clear. Therefore, I recommend adding a paragraph with the objectives (not the results) at the end of the introduction section. Moreover, the conclusions section should address the results concerning these objectives.

The figure captions should be self-explanatory, and the references should be added in the correct location of the text. In addition, authors must follow the journal's guidelines for manuscript writing. For example, sometimes the authors write “Fig” and others “Figure”. Please, unify. Moreover, some figures are not discussed or cited in the text (for example, figure 5). Moreover, in general, the quality of the figures should be improved and the figures should be explained in more depth in the text.

Finally, English needs a comprehensive revision all over the manuscript. Some mistakes must be fixed, such as grammatical errors or redundant verbs (some examples are indicated in the specific comments).

Therefore, I recommend a major revision of this manuscript.

Specific comments.

Line 18-22

Please, also include in the abstract the results regarding path length reduction.

Line 29

“Smart Agriculture is an important part of the smart economy”

Please, define “Smart agriculture” and “smart economy”, since they are confusing terms with different meanings depending on the context. Add references.

Line 37

“farmland obstacles pose a serious threat to agricultural UAVs.”

Please, add references to this statement. Moreover, what kind of obstacles? UAVs in agriculture are mainly used for monitoring crop status using cameras/spectral sensors. Therefore, UAVs fly over objects at a certain height and distance.

Line 49

“Anh et al.,”

Please, add the reference here.

Line 51

“To improve the efficiency of A-star algorithm in narrow spaces [16]”

It seems the fragment of a sentence.

Line 51-57

In general, this paragraph is a bit confusing and should be rewritten. Please be careful when inserting references.

Line 61

“Zhang et al.”

Please, add the reference here.

Line 71-77

This paragraph is a mixture between the Material and Methods section and the Results section. Therefore, I recommend moving this paragraph to these sections and substituting it with a paragraph showing the objectives of the paper.

Line 92

“during flight [.]”

Add a point.

Line 109-111

“The A-star algorithm will take significant time to compute, and the amount of storage extremely large, additionally the amount of computing time will determine the efficiency of routing.”

This sentence is hard to understand. Please, rephrase it.

Line 133

Please improve figure 4. It is a screenshot of an excel sheet (or other similar software).

Line 152

Where are figures 5a and b cited in the text?

Moreover, in figure 5A: Does the yellow dot supposed to be in the middle of the two cubes?

Line 156

Is this an image? Please, in order to improve resolution and quality, change it into text.

Line 158

“the path is not smooth enough”

What do you mean by “smooth enough”? straight? Please, specify.

Line 196

“Matlab”

Please, follow the guidelines for authors to reference these kinds of elements properly.

Line 213

Figure 8 is confusing. In this condition, it is hard to follow each path. Please, add three more views of this figure, showing only and specifically the XY, XZ, and YZ planes.

Line 238

Move the legend out of the figure.

Line 254

“We select scneraio”

Scneraio? Review the English language.

Line 290

“The RJA-star algorithm [is] mainly works offline on already known scenarios”

Please, delete “is” and check the English language.

Line 294

Delete one of the two points at the end of the sentence.

Reviewer 2 Report

The manuscript shows an improvement of previous published algorithm for UAV path optimization. Despite the fact that the topic is interesting and relevant, the document as such does not present important and significant improvements compared to the paper previously published by some of the authors in applied sciences journal 8.  https://doi.org/10.3390/app122211338. The introduction of the documents is very similar and some of the figures are the same ex. Fig 2, 3, and 4.   The literature review and conclusions need to be improved.   From my point of view, the general manuscript need to be rewritten to guarantee its originality.

Round 2

Reviewer 1 Report

The manuscript is interesting and was corrected scientifically according to some of the previous comments. However, as in the previous review, I recommend the authors to check the style, format and improve the text and figure captions since there are still errors. For example, in line 144 the authors write "Figure 4c" instead of "Figure 4b".

Moreover, a revision of the English language throughout the manuscript should be made.

Reviewer 2 Report

Important improvements were made to the manuscript in order to show their contribution and originality. However, it is important to review the connection with the new texts added. Additionally, Figure Two has two titles and its quality is low. The legends in figure 7 b and c are not easy to read. Table 2 and 3 should be on single page or repeat the table headings for ease reading. 
